# Assessing Protein Content and Dimer Formation in the Bevacizumab Reference Product and Biosimilar Versions Marketed in Spain

**DOI:** 10.3390/pharmaceutics16121520

**Published:** 2024-11-26

**Authors:** Alexis Oliva, Magdalena Echezarreta, Álvaro Santana-Mayor, Adrían Conde-Díaz, Joao Goncalves, Shein-Chung Chow, Matías Llabrés

**Affiliations:** 1Departamento de Ingeniería Química y Tecnología Farmacéutica, Facultad de Farmacia, Universidad de La Laguna, Avda. Fco. Sánchez, s/n, 38200 Santa Cruz de Tenerife, Spain; mechezar@ull.edu.es (M.E.); mllabres@ull.edu.es (M.L.); 2Servicio de Apoyo a la Investigación (SEGAI), Universidad de La Laguna, Avda. Fco. Sánchez, s/n, 38206 Santa Cruz de Tenerife, Spain; asantanm@ull.edu.es; 3Departamento de Química Analítica, Facultad de Ciencias, Universidad de La laguna, Avda. Fco. Sánchez, s/n, 38206 Santa Cruz de Tenerife, Spain; acondedi@ull.edu.es; 4Research Institute for Medicines (iMed.ULisboa), Faculdade de Farmácia, Universidade de Lisboa, Avenida Professor Gama Pinto, 1649-003 Lisboa, Portugal; jgoncalv@ff.ulisboa.pt; 5Department of Biostatistics and Bioinformatics, Duke University School of Medicine, Durham, NC 27710, USA; sheinchung.chow@duke.edu

**Keywords:** bevacizumab, biosimilar, analytical similarity, quality range method, between- and within-batch variability

## Abstract

Background: The manufacture of biologics is a complex, controlled, and reproducible process that results in a product that meets specifications. This should be based on data from batches used to demonstrate manufacturing consistency. Ten batches of originator product (Avastin^®^) were analyzed over a 10-year period. Methods: The β-expectation tolerance intervals and the process capability analysis were proposed to establish the specification limits for determining the acceptance criteria of the final product from the manufacturing process. Protein concentration and dimer content were utilized as CQAs. The analytical similarity between three biosimilars authorized in Spain since 2021 (Vegzelma^®^, Alymsys^®^, and Oyavas^®^) and the originator product were evaluated for both CQAs using two methods: the quality range (QR) method, based on one sample per batch, and the QRML one, which takes into account the inter- and intra-batch variability of the originator product. Results: The results indicate that the two main sources of variation are under control; even the level of variability observed is close to the capability of the analytical method. The manufacturing process, therefore, continues under statistical control. Similarity is demonstrated for the bevacizumab concentration regardless of the approach used, whereas similarity is demonstrated for the dimer content for only one of the biosimilar products. Conclusions: The proposed methodologies allow for the analysis of the consistency of the manufacturing process and the variability from batch to batch.

## 1. Introduction

A persistent upward trajectory in the number of biosimilar approvals has been documented in the global biopharmaceutical industry over the past decade. The availability of biosimilars at lower prices has the effect of reducing the cost to patients and healthcare and governmental systems, thereby increasing access to effective therapies. The manufacture of biological drugs is a complex process, which is sensitive to variations in the multi-step production process (expression system, cell growth conditions, purification process, formulation, and storage). This can result in the production of products that are heterogeneous. Consequently, the creation of biosimilars is an arduous undertaking, compounded by the dearth of information pertaining to the manufacturing process of the original product.

The quality of these products is primarily contingent upon the manufacturing process employed. The manufacturing process used to produce biological medicine is proprietary to the pharmaceutical company that developed it. Consequently, the biosimilar producer must develop its own manufacturing process, which may result in discrepancies between the biosimilar and its reference biological medicine. It is permissible for these differences to exist, provided that comparability studies demonstrate that the biosimilar has no clinically significant differences from its reference [1]. For approval following a comparability study, both the biosimilar and the originator product should contain the same active substance and be used at the same dose by the same route for the same indications [2]. It is possible that minor discrepancies may exist in the formulation (e.g., excipients), presentation (e.g., powder to be reconstituted/solution ready for injection), and administration device (e.g., type of delivery pen). However, these differences are not expected to impact the safety and efficacy of the final product [3].

From upstream production to the shelf, many parameters can affect the structure, function, immunogenicity, or stability of these products. For example, deamidation or oxidation can occur in almost all production steps, whereas glycosylation can be generated upstream [4]. In addition, small differences in the cell culture or the purification process can also lead to charge variants or facilitate aggregation [5,6]. Depending on their location, the binding kinetics of mAbs to their target and effector receptors can also be affected by various modifications [7,8].

Comparative quality studies rely on a wide range of analytical techniques for the comprehensive characterization of critical quality attributes (CQAs). Furthermore, this characterization is mandatory at different stages of product development, such as early development, stability studies, clinical trials, product approval, and post-marketing changes [9]. Therefore, during these processes, the differences between the originator and biosimilar products may become apparent.

Analytical similarity assessment is one of the critical steps for the success of biosimilar development, and manufacturers need to establish CQAs to ensure consistent product efficacy, safety, and quality [10]. The aggregate of therapeutic antibodies is usually considered to be one of the most important CQAs. Dimer formation is probably one of those CQAs that need to be monitored throughout the lifecycle of any given biological. The propensity of aggregate formation for bevacizumab is higher than other monoclonal antibody (mAb) drugs due to its tendency of self-association via the non-covalent interaction [11]. The results obtained by various authors indicated that the HMWS of bevacizumab was mainly composed of dimers [11,12]. Following regulatory agency guidance, biosimilars must demonstrate that the “strength”, measured as protein concentration, is the same as the reference product [13,14]. Zhang et al. [10] identified and classified the CQAs through risk assessment according to the impact of each quality attribute on efficacy and safety. In this context, the protein concentration and aggregate content are considered high-risk attributes.

Similarity acceptance criteria have been outlined and regulated by several authorities, including the FDA, EMA, WHO, and ICH, for the approval of newly manufactured biosimilars [15,16,17,18]. The FDA, assuming that the test product and reference product have similar population means and population standard deviations, suggests the use of QR methods for comparative analytical assessment [19]. In addition, they recommend the use of a minimum of 10 batches of reference products (purchased over a time frame that spans several years of expiry dates) and a minimum of 6 to 10 batches of the proposed product to meet the statistical criteria. On the other hand, because comparisons between population means (equivalence test) and between populations (QR method) are very different, the EMA even discourages the use of this approach to assess analytical similarity in biosimilar studies [20]. Nevertheless, various authors have analyzed from different points of view the limitations of both equivalence tests and QR approaches [19,20,21,22,23,24,25].

Son et al. [20] proposed two new versions of the QR method, which overcome the original’s limitations and allow detection of product changes during manufacturing.

However, the within-lot variability was not considered in this protocol. Recently, Oliva and Llabrés [26] analyzed the effect of within-lot variability on the QR. These authors proposed the estimation of QR bounds based on the variance components to account for both between-lot and within-lot variability, which was called the quality range maximum likelihood method (QR_ML_) to differentiate it from the currently used procedure based on one sample per batch.

It is important that the analytical and functional characterization of the biosimilar is carried out at the same time as that of the reference product, with validated methods to reduce variation and accurately determine the extent of the biosimilarity.

Bevacizumab is a humanized anti-VEGF (vascular endothelial growth factor) antibody that inhibits tumor vascularization by binding to the VEGF, an angiogenesis-inducing growth factor. Bevacizumab, known commercially as Avastin^®^ (Genentech Inc., Roche), was first approved by the FDA in 2004 and by the EMA in 2005 as a first-line treatment for colorectal cancer [27,28], but it is approved for many other cancers [29,30,31].

Due to the growing demand in this area, many pharmaceutical companies are developing bevacizumab biosimilars. More than ten biosimilar candidates are being developed, and several biosimilars are approved by the FDA, EMA, or other national regulatory authorities [32]. Currently, in Spain, there are eight authorized and marketed biosimilar products since 2019 (Mvasy^®^ and Zyrabev^®^ were the first in September 2019, and the last was Abevmy^®^ in February 2024 [33]).

In this context, this study had three principal objectives: first, to ascertain whether the manufacturing process for the reference product is under statistical quality control over a period of 10 years, and, second, to determine whether the aforementioned process is capable of maintaining the requisite quality standards. To this end, β-expectation tolerance intervals were calculated using variance components in order to account for both between-batch and within-batch variability. In addition, a process capability analysis was conducted through the Cpk capability index. Third, analytical similarity was assessed using two methods based on the quality range (QR) approach: the QR method, as proposed by the FDA, and the QR_ML_ method, as proposed by Oliva and Llabres [26]. The former involves each batch providing one test value for the CQAs being assessed, while the latter is an alternative approach to testing multiple samples from each batch in order to account for potential scenarios and ensure fair and reliable comparisons are made. In this study, the protein concentration and dimer content were utilized as CQAs to evaluate the analytical similarity between three authorized biosimilars in Spain since 2021 (Vegzelma^®^, Alymsys^®^, and Oyavas^®^) and the originator product (Avastin^®^). A robust SEC method was used to evaluate both CQAs. In addition, all samples were analyzed under the same conditions to ensure a reliable comparison of all these products.

## 2. Materials and Methods

### 2.1. Materials

Ten batches of bevacizumab originator (Avastin^®^, Roche, Munich, Germany) were chosen as the reference product and tested during a period of 10 years. One batch of three biosimilar drugs, authorized and marketed in Spain since 2021, were used as bevacizumab biosimilars (Alymsys^®^, GH GenHelix SA, León, Spain; Oyavas^®^, Stada Arzneimittel AG, Bad Vilbel, Germany; Vegzelma^®^, Millmount Healthcare Ltd., Versegyhaz, Hungary). Alymsys^®^ and Olaya^®^ are produced by the same manufacturer but with a different shelf life. This legally is correct. All brands were procured from licensed vendors as a 25 mg/mL solution in phosphate buffer and consisted of trehalose dihydrate, sodic phosphate, and polysorbate 20.

Different numbers of replicates were used for all experiments conducted within the shelf life of the products and tests. The biosimilars were named biosimilar #1, biosimilar #2, and biosimilar #3 in the same order as above.

Deionized water was purified in a MilliQ plus system from Millipore (Molsheim, France) prior to use. All other chemicals and reagents were HPLC grade. All solvents were filtered with 0.45 µm (pore size) filters (Millipore) and degassed.

### 2.2. Size Exclusion Chromatography System

The chromatographic apparatus used was a Waters system (Milford, MA, USA) comprising a pump (600E Multisolvent Delivery System), an autosampler (model 700 Wisp), and a differential refractive index (RI) detector (Waters model 2414). Elution was performed at room temperature on a Protein KW-804 column (8 × 300 mm, Waters). The mobile phase was phosphate-buffered saline containing 0.3 M NaCl, 0.025 M phosphate, and pH 7.0. The flow rate was set at 1.0 mL/min, and the injection volume was set at 25 µL. Waters Millennium 32^®^ chromatography software (version 3.2) was used to collect and analyze the data. The software was used for the integration of the chromatograms and the estimation of the monomer and dimer content by the calculation of the relative percentages of the peak areas. Appendix A shows a representative chromatogram for each product tested.

### 2.3. Analytical Method Validation

The proposed analytical method was validated and demonstrated as suitable for the proposed application. In this process, the protein content, expressed as a percentage of the actual concentration in relation to the claim value, and the dimer content (expressed as a percentage) were used as CQAs. Briefly, linear calibration was performed with six independent assays, and each assay included five independent samples with bevacizumab concentrations ranging from 5 to 30 g/mL (n = 30). The statistical model included concentration, day (as a factor), and the interaction concentration by day coefficients. A simple linear model (*p* > 0.05) was used because the null hypothesis for day and interaction concentration by day was accepted. This assumes that data from different days can be pooled to obtain the regression line and that the observed variability is only due to an error in the analytical method. In our case, a linear relationship was found between bevacizumab peak areas (µVs) and corresponding concentrations (µg/mL). The plot of reference versus predicted values for the calibration samples was used to verify the absence of systematic bias. A similar result was obtained for dimer content. For further details, see Oliva and Llabres [34]. This analytical method was re-validated during the conduct of this study, with similar results and confirmation of suitability for the intended use.

### 2.4. Statistical Model

The statistical model used is
(1)$$y_{\left(ij\right)}=\mu +B_{i}+\varepsilon_{\left(ij\right)}$$
where y_ij_ is the observation j (j = 1, …, ni) from batch i (i = 1, …, m), µ is the general mean, B_i_ is the batch random effect, and e_ij_ is the residual random term accounting for sampling variability and analytical method uncertainty. The random terms B_i_ and e_ij_ are assumed to be independent and with distribution N (0, $\sigma_{B}^{2}$) and N (0, $\sigma^{2}$), respectively.

The variance components of the statistical model (see Equation (1)) were estimated using the maximum likelihood method and its application in unbalanced designs [34]. This was achieved through the use of the function lmer() from the “R” program (version 4.4.1) [35]. The β-expectation tolerance intervals were calculated using the approach outlined by Hoffman and Kringle [36].

## 3. Results and Discussion

### 3.1. Manufacturing Process Control

#### 3.1.1. Reference Product

Ten batches from Avastin^®^ (reference product) were analyzed during a period of 10 years. Six batches were analyzed during the period 2015–2021 (for more details, see Oliva and Llabres, 2021) [34]; a further four batches were tested in the period 2022–2024 (one batch in 2022 and the rest during the first semester of 2024), and there were ten batches in total. Table 1 shows the summarized statistics: number of observations per batch and the mean and standard deviation for both quality attributes included in this study (all data are available in the Appendix A). For the 10 batches, overall bevacizumab mean content was 24.82 mg/mL; within-batch standard deviation ranged from 0.035 to 0.209 (coefficient of variation from 0.14 to 0.84%). Overall bevacizumab mean dimer content was 1.559%, and the within-batch standard deviation ranged from 0.034 to 0.195 (coefficient of variation from 2.30 to 13.9%). In this case, the average within-batch variability, expressed as a coefficient of variation (CV), for the drug content was lower than those obtained for the first six batches (0.40% vs. 0.55%) [34], whereas the percent dimer content was similar (8.71% vs. 8.77%), although the problems in the resolution between peaks, detected in the first period [34], were also observed in various batches, reaching a CV close to 14% for batches #8 and #10. However, the last four batches show the lowest CV for the drug content (<0.21%).

Table 2 shows the variance components estimated by the maximum likelihood method with their 95% confidence intervals. The obtained results show that the null hypothesis, $H_{0}:\sigma_{B}^{2}=0$, for the percent dimer content is accepted because the confidence intervals include the zero value. Therefore, there are no differences between batches and the observed variability is due to within-batch factors. These are mainly analytical method uncertainty and sampling variability, with an overall CV of 9.23%. The same result and conclusions were obtained in the first study.

For the protein concentration attribute, the null hypothesis is rejected, $H_{0}:\sigma_{B}^{2}\neq 0$, and, therefore, there are differences between batches. In this case, the between-batch variability was higher than the within-bath variability (i.e., analytic method uncertainty), and the total variability, expressed as the CV, was 1.0%. Similar results were obtained in the first study. Thus, all these data confirm that the batch-to-batch variability is not negligible because of the lack of published manufacturing records. In addition, the manufacturing process of biopharmaceuticals is a complex process with a rigorous quality control system, but this process occurs with a certain degree of variability.

For example, in the assessment report of the European Medicines Agency (EMA) for the ABP 215 biosimilar, the sponsor indicated that one batch had a purity profile that was just below the range of comparability presented. However, the ABP 215 product is analytically similar to the reference product, and this small discrepancy is clinically insignificant [37].

In a previous study, Oliva and Llabrés [34] proposed the application of β-expectation tolerance intervals as specification limits for the determination of the final product’s acceptance criteria from the manufacturing process. If a future observation is expected to be produced from a new batch, it must fall within these limits in order to ensure its quality. Figure 1 shows the individual observations for each reference product batch, together with β-expectation tolerance intervals computed using the proposed approach by Hoffman and Kringle for a design with unequal numbers of repeated measures in each batch. For more details, see Oliva and Llabrés, 2021 [34].

In a such scenario, all data points for the new reference product batches were within the estimated tolerance intervals for both CQAs. Therefore, the manufacturing process continued under statistical control. However, the mean bevacizumab content exhibited a slight shift from 24.89 mg/mL to 24.82 mg/mL, accompanied by a narrowing of the β-tolerance intervals. This finding may be attributed to the reduced sample size and diminished experimental variability observed for the four subsequent test batches.

The process capability to keep the manufacturing under statistical quality control must be assessed and continuously improved after approval as part of product life cycle management. In this process, the Cpk capability index was calculated. For instance, the lower and upper specification interval, [LSL, USL], must be set to provide assurance that the drug concentration will be within the specification limits until the drug product´s expiration time. LSL and USL were calculated as 95 and 105% of the labeled bevacizumab concentration [23.75, 26.25], whereas the lower and upper capability limits (LCL, USL) were computed from the parameter model shown in Table 2 based on the µ ± 3σ rule for ten batches [16]. In this context, calculated LCL and LSL were 24.16 and 25.48 mg/mL, respectively. The Cpk estimated by the bootstrap method is 1.62. Its 95% confidence interval is 1.19–2.30. All these data, which exceed the 4 σ level commonly used in pharmaceutical manufacturing, confirm that the process is not only of statistical quality but also meets the specification of drug content.

#### 3.1.2. Biosimilar Products

In this work, we tested three bevacizumab biosimilar commercial products manufactured at different times and showing at least 1-year residual shelf lives from six measurements for each batch. Figure 2 shows the plot of the observation by different biosimilar products together with the lower and upper capability limits as well as the β-tolerance intervals calculated for the reference product. In such a scenario, all the biosimilar products comply with the drug content specifications. However, the mean values for biosimilar products #1 and #3 were below the target value and the mean reference product content, whereas biosimilar #1 was very close to the reference product value. However, the situation was totally different for the dimer content, with ranges from 0.808 to 2.083%, whereas the mean value for the reference product was 1.559%. This could be attributed to the distinct manufacturing processes employed for each biosimilar promotor. Biosimilar #1 and #2, which were manufactured using the same manufacturing process, showed differences in dimer content. However, we do not have any information on the level of variability to establish a significant difference. This is an industrial secret. We need more data to determine if there really are differences between batches of the same promotor.

Due to the unique biosynthetic manufacturing process and molecular characteristics of biotechnology products, the drug substance may contain multiple molecular entities or variants. If these molecular entities are derived from the expected post-translational modification, they are part of the intended product. If variants of the intended product are formed during the manufacturing process and/or storage and have properties similar to those of the intended product, they are considered to be product-related substances and not impurities [12]. In this context, individual and/or collective acceptance criteria for product-related substances should be set, as appropriate. Using the β-tolerance intervals as specification limits (or acceptance criteria) may be an alternative, but in our case, two of the biosimilar drugs analyzed show levels outside this interval (see Figure 2). The best solution may be to set a maximum value, as is performed for impurities.

### 3.2. Analytical Similarity

In the last guidance published in 2019, the FDA recommends a quality range (QR) method in the comparative analytical assessment [15]. In addition, in the section “Considerations for Reference and Biosimilar Products”, the Agency (FDA) recommends that a sponsor include at least 10 reference product lots (acquired over a time frame that spans expiration dates of several years), whereas for the proposed product “the Agency recommends that a sponsor include at least 6 to 10 lots of the proposed product” in the comparative analytical assessment.

The QR of the reference product for specific CQAs is defined as follows:$${(\hat{\mu }}_{R}-k\cdot {\hat{\sigma }}_{R},\;{\;\hat{\mu }}_{R}+k\cdot {\hat{\sigma }}_{R})$$
where ${\hat{\mu }}_{R}$ is the reference product lots mean, ${\hat{\sigma }}_{R}$ is the standard deviation of the reference product lots, and k (symbol x in the original FDA document) should be appropriately justified. A comparative analysis of a quality attribute would generally support a finding that the proposed product is highly similar to the reference product when a sufficient percentage of biosimilar lot values (e.g., 90%) fall within the QR defined for that attribute [15].

The QR method is applied unconditionally, but it is necessary to fix the k constant value. Recent publications and FDA presentations specify k values ranging from 2 to 3 to assure that there will be a high percentage (e.g., at least 95%) of the test values falling within the QR [21,22,24,38,39].

Oliva and Llabrés [21] analyzed the effect of the selection of the k value on the probability of passing the similarity test in the function of mean shifts and relative variability of the tested product and the reference product. The authors indicate that the pass rate is higher than 90% for k = 3 and lower than 50% for k = 2 for those scenarios with small variabilities (σT/σR < 0.6), and the mean shift is lower than 4%.

Table 3 shows the QR_ML_ intervals and 95% confidence intervals for both lower and upper limits considering 10 reference product batches and for k = 3. For bevacizumab concentration, expressed as a percentage of the claim value, the estimated QR_ML_ was [97.08, 101.51], and the 95% confidence intervals were 96.64 and 97.53 for the lower bound and 101.07 and 101.95 for the upper bound. For dimer content (%), the estimated QR_ML_ was 1.174 and 1.883, and the 95% confidence intervals were 1.103 and 1.245 for the lower bound and 1.812 and 1.95 for the upper bound.

To show that the $QR_{ML}$ method provides more reliable results than those given by the FDA proposed approach (one sample per batch and 10 batches), we computed the expected distribution of the latter using a stratified bootstrap sampling (one sample per batch; 2000 simulations) from the experimental data (see Table 4). The obtained results for both parameters were very similar, although the 95% confidence interval was slightly broader.

To apply the QR method, and for illustration purposes, we arbitrarily assigned a batch number to each sample, obtaining data from six batches of each proposed product.

Table 5 shows the bevacizumab content and percent dimer content for each biosimilar product.

Biosimilar #2 and #3 present values for both CQAs that are slightly different with respect to the reference product; they highlight the difference in bevacizumab content between the brands. A difference of one percentage point was observed. However, biosimilar #2 has a content of bevacizumab similar to the mean value for the reference product (99.22% vs. 99.29%). On the other hand, these differences are significant, which is especially true for the dimer content, where this one was half of the estimated value for the reference product in the case of biosimilar #2, whereas it was approximately 1.5 times higher for biosimilar #3. Thus, this finding could be due to its own manufacturing process, perhaps in the purification step, although the aggregation can take place in filling vials, leading to a larger sample-to-sample variability, which may explain the discrepancies between the different biosimilar products and the originator. It is important to note that biosimilar #1 and #2, produced by the same manufacturer but marketed by two different companies, should be considered in relation to any potential issues that may arise from their distribution or storage conditions.

Figure 3 shows the obtained results in the comparative analytical assessment for two CQAs using both QR and QR_ML_ approaches and using k = 3. For the bevacizumab content CQA, the similarity is demonstrated for all the biosimilar brands since more than 90% of the individual values (all points) fall within the QR of the reference product, independently of the applied approaches (Figure 3A). This situation corresponds to scenarios where the mean shift is smaller (±1%) and the variability ratio is lower than 0.6. Thus, the probability of passing the similarity test is higher than 90% [16]. However, the situation is completely different for the dimer content CQA (see Figure 3B), where the similarity is only demonstrated for biosimilar #1 using both approaches. In the case of biosimilar #3, the probability of passing the test is lower than 17.6%, whereas it is zero for biosimilar #2, independently of the applied approaches. This situation corresponds to a scenario with a relatively large mean shift (>5%), where the similarity is not demonstrated independently of the k value or variability considered [21].

## 4. Conclusions

A certain degree of heterogeneity between batches of biological products is expected due to the unique biosynthetic process used to produce them. This heterogeneity may occur during the manufacture and/or storage of these products and defines their quality. The manufacturer should demonstrate and ensure batch-to-batch consistency. In this context, the combination of β-expectation tolerance intervals and process capability limits allows specification limits to be set based on data obtained from batches produced over a 10-year period, and the consistency of the manufacturing process and batch-to-batch variability to be assessed.

The analytical similarity between biosimilar products and originators was assessed using two approaches based on the quality range (QR). The QR_ML_ method shows several advantages over the FDA-proposed one. First, variance components from the statistical model allow the type of variability of the reference product to be analyzed. In such a scenario, the batch-to-batch variability is not negligible, whereas the relevance of the within-batch variability (i.e., analytical method error) can be anticipated from the validation procedure. Second, this helps to calculate more accurate QR values, as well as set a priori values for the QR. This study demonstrated that the three biosimilar products are highly similar to the originator in terms of protein concentration, regardless of the approach used, whereas the dimer content, is only demonstrated for biosimilar #1. However, all these results and conclusions need to be confirmed with real data from 6 to 10 batches of biosimilar products.

## Figures and Tables

**Figure 1 pharmaceutics-16-01520-f001:**
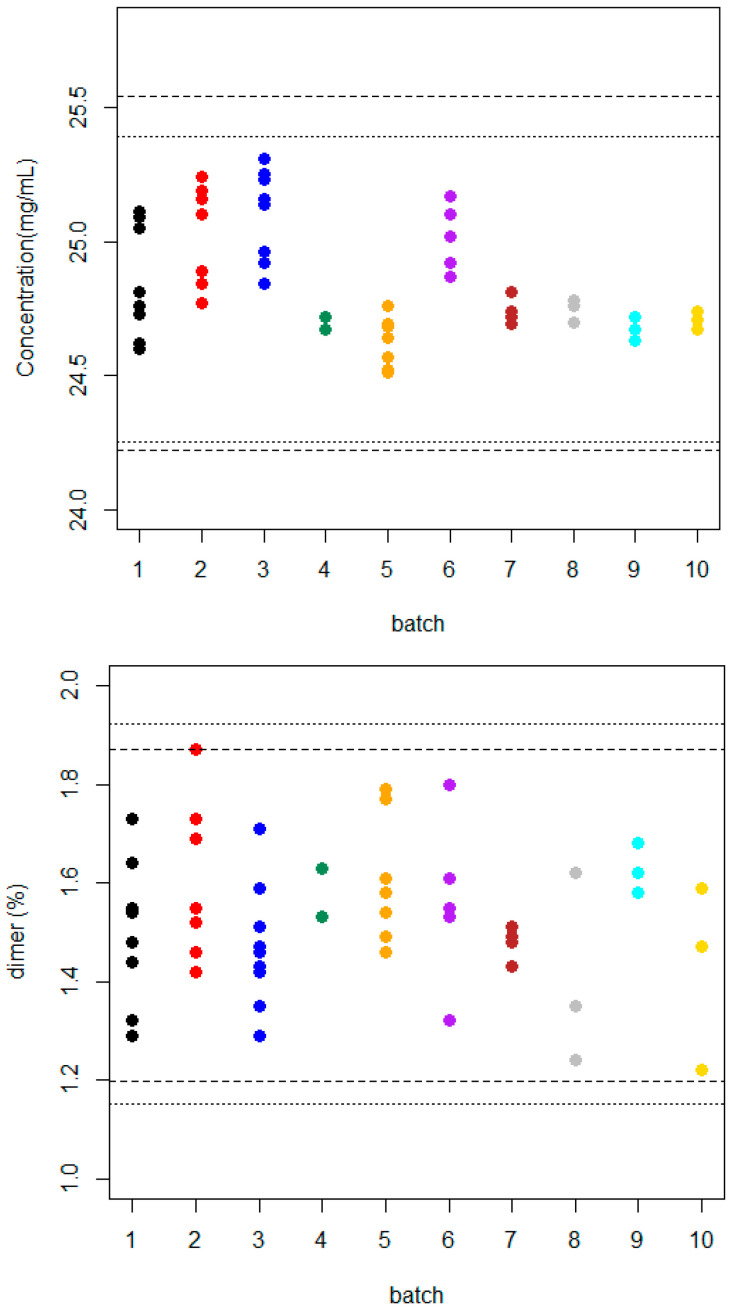
Observed data sets for ten Avastin^®^ batches and β-expectation tolerance intervals for both CQAs (upper, bevacizumab concentration; lower, percentage dimer content) calculated using six (----) batches taken from reference [34] and ten (….) batches in total data.

**Figure 2 pharmaceutics-16-01520-f002:**
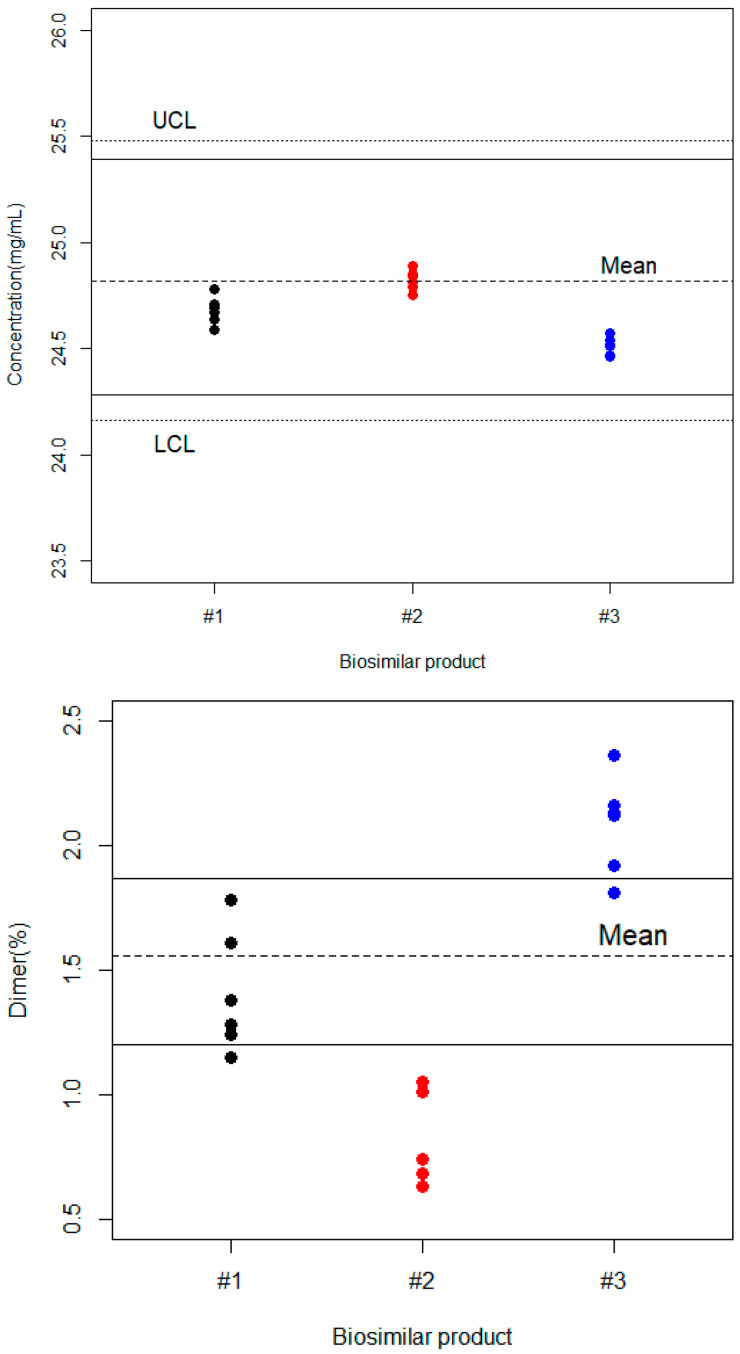
Observed data set for the three biosimilar products together with the lower and upper capability limits (dotted lines) and β-expectation tolerance intervals estimated for the reference product (continuous lines).

**Figure 3 pharmaceutics-16-01520-f003:**
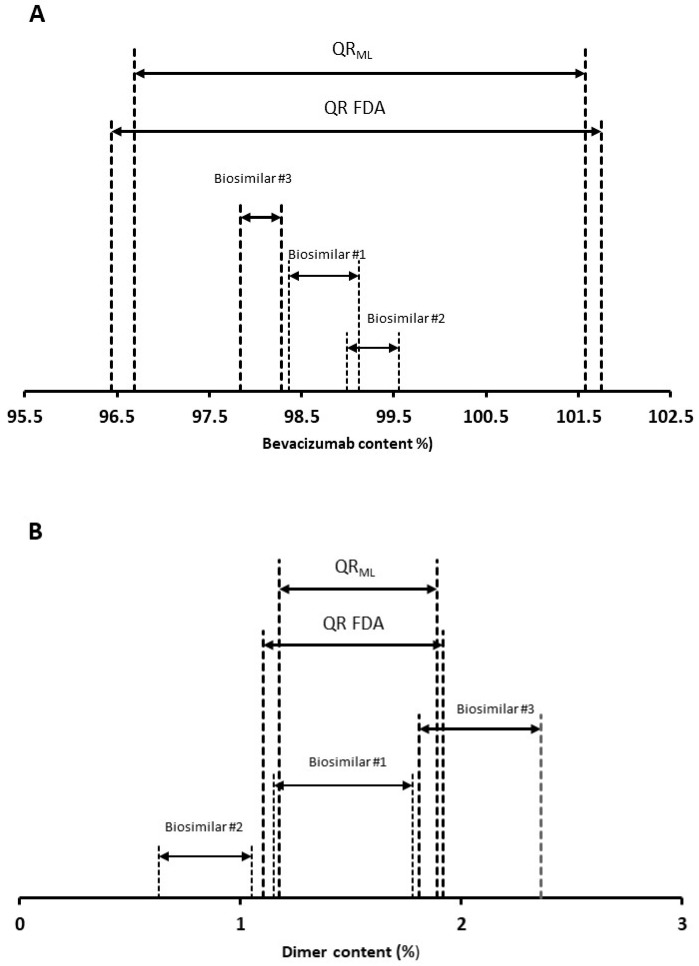
Results of the comparative analytical assessment for both CQAs using the QR and QR_ML_ approaches. (**A**) Bevacizumab content (%) and (**B**) percent dimer content.

**Table 1 pharmaceutics-16-01520-t001:** Summary statistics of experimental data (sample size, mean, standard deviation, and coefficient of variation (CV%)) for both CQAs and batches from reference products used in this study. Data from batches 1 to 6 were taken from reference [34].

		Content (mg/mL)			Dimer (%)		
Batch	Sample Size	Mean	SD	CV (%)	Mean	SD	CV (%)
1	8	24.85	0.209	0.84	1.499	0.150	10.01
2	7	25.03	0.189	0.76	1.606	0.162	10.09
3	9	25.11	0.167	0.67	1.470	0.125	8.50
4	2	24.69	0.035	0.14	1.580	0.071	4.49
5	8	24.61	0.096	0.39	1.587	0.130	8.19
6	5	25.02	0.124	0.50	1.562	0.172	11.01
7	4	24.74	0.051	0.21	1.478	0.034	2.30
8	3	24.75	0.042	0.17	1.403	0.195	13.90
9	3	24.67	0.045	0.18	1.621	0.050	3.08
10	3	24.71	0.035	0.14	1.427	0.189	13.24

**Table 2 pharmaceutics-16-01520-t002:** Maximum likelihood method estimates (QR_ML_) and their corresponding 95% confidence intervals for the first six batches (upper) and the ten batches in total (lower) of the reference product. Data from batches 1 to 6 were taken from reference [34].

6 Batches				
CQA	Parameter	Estimation	Lower Bound	Upper Bound
Content (mg/mL)	${\hat{\sigma }}_{B}$	0.190	0.093	0.364
	$\hat{\sigma }$	0.162	0.130	0.210
	$\hat{\mu }$	24.89	24.71	25.06
Dimer (%)	${\hat{\sigma }}_{B}$	0.029	0.000	0.098
	$\hat{\sigma }$	0.142	0.114	0.181
	$\hat{\mu }$	1.543	1.490	1.600
**10 Batches**				
**CQA**	**Parameter**	**Estimation**	**Lower Bound**	**Upper Bound**
Content (mg/mL)	${\hat{\sigma }}_{B}$	0.167	0.098	0.276
	$\hat{\sigma }$	0.145	0.119	0.182
	$\hat{\mu }$	24.82	24.70	24.94
Dimer (%)	${\hat{\sigma }}_{B}$	0.034	0.000	0.096
	$\hat{\sigma }$	0.141	0.116	0.176
	$\hat{\mu }$	1.523	1.478	1.573

**Table 3 pharmaceutics-16-01520-t003:** Estimates and 95% confidence intervals for the QRML estimated from variance components and number of batches. Data from batches 1 to 6 were taken from reference [34].

6 Batches				
	QR Bounds	Lower	Estimate	Upper
Bevacizumab content (%)	lower	96.56	97.06	97.56
	upper	101.56	102.06	102.56
Dimer (%)	lower	1.118	1.193	1.268
	upper	1.866	1.951	2016
**10 Batches**				
Bevacizumab content (%)	lower	96.64	97.08	97.53
	upper	101.07	101.51	101.95
Dimer (%)	lower	1.103	1.174	1.245
	upper	1.812	1.883	1.955

**Table 4 pharmaceutics-16-01520-t004:** Estimates and bootstrap 95% confidence intervals for the QR obtained for the simulation data set and number of batches. Data from batches 1 to 6 were taken from reference [34].

6 Batches				
	QR Bounds	Lower	Estimate	Upper
Bevacizumab content (%)	lower	95.91	96.69	97.47
	upper	101.22	102.28	103.35
Dimer (%)	lower	0.972	1.172	1.372
	upper	1.742	1.972	2.203
**10 Batches**				
Bevacizumab content (%)	lower	96.30	96.91	97.52
	upper	100.65	101.56	102.48
Dimer (%)	lower	0.978	1.112	1.246
	upper	1.777	1.917	2.057

**Table 5 pharmaceutics-16-01520-t005:** Bevacizumab and dimer content, expressed as a percentage, for each biosimilar product.

	Bevacizumab (%)			Dimer (%)		
Product	Mean	Sigma	Min–Max	Mean	Sigma	Min–Max
Biosimilar #1	98.72	0.259	98.36–99.12	1.405	0.242	1.145–1.780
Biosimilar #2	99.22	0.220	98.99–99.56	0.808	0.176	0.630–1.048
Biosimilar #3	98.17	0.270	97.84–98.52	2.083	0.195	1.808–2.356

## Data Availability

Data are available within this article and the associated Appendix A.

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
