# Peer review of "Assessing Protein Content and Dimer Formation in the Bevacizumab Reference Product and Biosimilar Versions Marketed in Spain"

_pharmaceutics, 2024, doi:10.3390/pharmaceutics16121520_

Round 1
Reviewer 1 Report (Previous Reviewer 1)
Comments and Suggestions for Authors
Please, see attached file. Although I suggest refinements, the paper has greatly improved. Thanks

Author Response
Manuscript ID: Pharmaceuticals_3324279_ revised version
Replies to reviewer´s comments
Reviewer #1
Pharmaceutics Ms: 3324279 -revised version
“Assessing Analytical Similarity Between Three Biosimilar Bearing Bevacizumab Marketed in Spain and the originator product”
(This is a new submission after revising the previous manuscript “Assessing Analytical Similarity of Three Bevacizumab Biosimilar Marketed in Spain to Avastin®” (Ms 3260259))
General Comments
I thank the authors for their effort in reshaping the manuscript.
They note that my previous review was unclear when referring to “The paper would have been of relevance had the study been executed differently”. The authors claim that “The reviewer doesn't say how this could be done. So, it is very difficult for these authors to provide an alternative text.” Well, what comes after that sentence in my first Revision Note are some of the” different executions” I am suggesting. So, they have actually addressed it. No need to further into this.
Thanks for your comments
I am not sure that, as the authors claim “drift” does not refer to any quality attribute. Dimer formation is probably one of those quality attributes that need to be monitored throughout the lifecycle of any given biological. However, if I am mistaken, which I may, I’d suggest the authors to make that concept clear in the paper.
Thanks for your suggestion. This was added in the new revised version
- - I am afraid that, for the purpose of this study, I can’t agree on the point that Oyavas and Alymsys are 2 different biosimilars. Given that we know that the same manufacturer produces them, it does not seem right to establish the same “distance or relation” between “Vegzelma and Oyavas” as between “Alymsys and Oyavas”. The authors have just added a sentence referring to the legal side but that is not enough. “Vegzelma and Oyavas” are 2 different biosimilars of the same reference medicine Avastin as we understand it, but “Alymsys and Oyavas” should be treated different given that only the name changes.
- - I believe that the comment on the legal discrimination is not what I meant. Of course, it is legally correct that a manufacturer produced for 2 MAH under different brand names. I am not discussing that. What I question is that (1) for the purpose of your paper you should consider them 2 different biosimilars and (2) I would discuss the fact that you find differences between Oyavas and Alymsys which factually are the same product. The authors provide an explanation as to why then differences exist that may well be true (different bacthes), but, may there be other explanations? Issues with distribution or storage of the same product by different MAH?
This is true, but legally there are three biosimilars authorized in Spain.
From the manufacturer's point of view, the situation is different, as Oyavas® and Alymsys® are produced by the same manufacturer and the same product, but in different batches. Initially, the data seems to indicate that there are differences between them in the dimer content, although the level of variation between batches is unknown (If we assumed two different batches from the same manufacturer, it is impossible to establish significance statistical difference with two points).
We do not have any information on the level of variability to establish a significant difference. This is an industrial secret. We need more data to determine if there really are differences between batches of the same manufacturer.
In this context, the dimer content is calculated by the relative peak area percentage as a result of integration. The variation is close to 9%. However, this value is not sufficient to explain this difference. There may be other reasons, as suggested by the reviewer: issues related to distribution or storage of the same product by different MAH, which should be considered. In our opinion, more data are needed to confirm this conclusion.
All these issues were included in the text.
- I am sure about what the authors mean by “However, the original idea was to study the stability of bevacizumab and not biosimilarity” and why do they say so. Just in case that it addresses that specific comment, I am not questioning the use of “protein content” and “loss of monomers” as important attributes, I am just asking to (1) be clearer on the attributes analysed, and (2) the reasons behind such specific analysis. The current version is certainly clearer to that respect.
Thanks for your comments.
- The phrase was modified “The biosimilar can be administered in the same manner and via the same route as the reference medicine, although the subcutaneous infliximab biosimilar “Remsima SQ” is an exception” is not correct from a regulatory perspective. Rather than a exception, it is the only one having evolved towards a different route of administration. Maybe you could say “For approval after a comparability exercise, both biosimilar and the original reference medicine should bear the same active substance, be used at the same dose through the same route and in the same indications
Thanks for your suggestion. The wording has been changed as you suggested.
- Genentech was not correctly written…Now it is. Thanks
- Which are biosimilar # 1, 2 and 3 should be said.
This information was included in Materials and Methods section.
- I still think that the title is misleading. You are not assessing analytical similarity, which would imply many more tests..your measuring two specific relevant quality attributes. Why don’t you actually emphasize that in the title. For instance, “Assessing protein content and dimer formation in the original medicine bearing bevacizumab and biosimilar versions marketed in Spain” (you may reduce/change). To that extent saying throughout the text that “the analytical similarity between biosimilars products and originator was assessed.” is misleading…it is the similarity in 2 attributes rather than “analytical similarity”, in my view.
Thanks for your suggestion. The “analytical similarity” term imply more tests and we used only two specific and relevant quality attributes. The title was changed as you suggested with minor changes. In addition, we deleted the number of biosimilar products to avoid confusions.
The new title: Assessing protein content and dimer formation in the bevacizumab reference product and biosimilar versions marketed in Spain.
Conclusion “The study demonstrated that the three biosimilar products are highly similar to originator in terms of protein concentration, regardless of the approach used, whereas for the dimer content it is only demonstrated for the biosimilar #1. However, all these results and conclusions need to be confirmed with real data from 6 to 10 batches of biosimilar products”. Again, how would you explain that the same product marketed by two MAH exhibits those differences if that is the case.
The reasons for these differences are many and varied, and the reviewer has before mentioned some of them.
We think it's a good question, but at the moment and with the data and information available, it's not possible to give an unique explanation. With more data, we could estimate the variability between batches to establish a significant difference. In such a scenario, using the β-tolerance intervals as specification limits could be an alternative, but in our case a possible solution could be to set a maximum value as an acceptance criterion, as is done for impurities. This was commented in the original and revised version. The following question is: what is the maximum value? This issue will be the subject of future research.

Reviewer 2 Report (Previous Reviewer 2)
Comments and Suggestions for Authors
none
Author Response
Thanks
Reviewer 3 Report (Previous Reviewer 3)
Comments and Suggestions for Authors
The authors have improved the manuscript according to the reviewers questions. I keep my minor comment that a representative chromatogram should be included in supplementary material.
Author Response
Reviewer #3
Thanks for your suggestion. A representative chromatogram of the different products has been included in the supplementary material as Figure S1. In addition, the data file from Excel is available.
Round 2
Reviewer 1 Report (Previous Reviewer 1)
Comments and Suggestions for Authors
See attached file

Round 3
Reviewer 1 Report (Previous Reviewer 1)
Comments and Suggestions for Authors
Ready for publication as far as I am cancerned. Thank you for addressing my comments, and for your patience.
This manuscript is a resubmission of an earlier submission. The following is a list of the peer review reports and author responses from that submission.
Round 1
Reviewer 1 Report
Comments and Suggestions for Authors
I am afraid that the manuscript is difficult to follow because of the various facets. This reviewers recommends to focus on one of the objectives mentioned and deepen into it. On the other hand I find serious issues conveyed i my revision note.

Author Response
Reviewer #1
Pharmaceutics Ms: 1779557
“Assessing Analytical Similarity of Three Bevacizumab Biosimilar Marketed in Spain to Avastin®”
The authors analyse batches of the original product bearing bevacizumab, Avastin. They check for consistency of quality attributes during a 10 year-period. Based on that analysis of the reference medicine they assess whether biosimilars of Avastin sourced from the Spanish market meet the specifications and are sustainably within the “biosimilarity range”. They conclude that variability is well in control no matter the analytical approach (whether comparability on the basis of QR or population mean is used).
General Comments
- The idea behind that paper is of interest since it addresses the concept of the “drift” of marketed biologics (although it normally applies to structural changes rather than content issues). The issue under debate is whether such drift throughout their lifecycle may create divergences between biosimilars and the original reference product. The paper would have been of relevance had the study been executed differently. (surprisingly the term “drift” does not appear in the manuscript).
We appreciate the reviewers comment and valuable suggestions. In this context, the term “drift” does not appear in the manuscript since it applies to structural changes rather than content issues. This fact was not analyzed in our study.
With respect to the comments: “The paper would have been of relevance had the study been executed differently”. The reviewer doesn't say how this could be done. So, it is very difficult for these authors to provide an alternative text.
The are 4 major issues:
- - One relevant issue is the selection of the quality attributes. Why specifically bevazicumab mean content and bevacizumab mean dimer content? Is there any hypothesis behind the expected differences? Are in-product or bevacizumab versions known for dissimilarities at those levels? Aren’t there many other attributes that would possibly be affected throughout any product’s lifecycle (glycopattern?). There may be a justification for this specific study but should be very clearly stated and explain in the introduction.
- We appreciate the reviewer’s comments.
- In 2015-2016, we started working on the bevacizumab drug. Our first study used size exclusion chromatography with light scattering detection (SEC-MALLS) to characterize the type of bevacizumab aggregates that form under mechanical and thermal stress, and to monitor their aggregation (Oliva et al.; Eur J Pharm Sci, 2015, 77, 170).
- In many of kinetics models, the experimentally observed kinetic are often quantified with respect to monomer concentration, fraction of monomers converted to aggregate (such as dimer, trimer, etc) or loss of monomer at a given time. In this context, SEC is a sufficiently robust analytical method that provides a convenient means to simultaneously generate kinetic data and aggregate morphology. In a second study, this analytical method was validated using a pre-study validation process in accordance with the ICH-Q2 (R1) guidelines and in-study monitoring in accordance with the Analytical Target Profile (ATP) criteria. The total error and β-expectation tolerance interval rules were used to assess method suitability and control the risk of incorrectly accepting unsuitable analytical methods (Oliva et al., J. Chromatogr B, 2016, 1022, 206). Later, in 2019, this method was applied to the analytical similarity assessment based on the QR method using real data from various bevacizumab lots (Oliva & Llabrés, Separations, 2019, 6, 43). The protein content, expressed as a percentage of the actual concentration relative to the claimed value, was used as a high-risk attribute.
- It is true, there are other attributes that would be analyzed. However, the original idea was to study the stability of bevacizumab and not biosimilarity. The latter has been developed by our research team since 2019, following recommendations from regulatory agencies (https://www.fda.gov/media/125484/download). In addition, many authors have used protein content as a high-risk attribute and we want to assess analytical biosimilarity from this point of view, focusing about these two attributes.
- -Another very relevant point is that Alymsys® and Oyavas® are considered by the authors as 2 biosimilars of Avastin. While legally it is the case, the fact is that they are both produced by the same manufacturer and therefore are the same product with 2 brand names (market by different MAH). Hence, rather than 3, only two biosimilars sourced from the Spanish market have been studied. This should be acknowledged in the manuscript. It is not correct for the purpose of this paper to discriminate between Alymsys and Oyavas. Interestingly in spite of being the same products they appear to exhibit significant differences. Given we know that they come from the same manufacturer, how should this be interpreted?
Thanks for your information. It is true, both biosimilar are market by different MAH and produced by the same manufacturer. Based on this information, they should not exhibit significant differences if the manufacturing process was the same, although this process can occur with a certain degree of variability. However, the shelf life was different. We could therefore assume that the batches were produced at different times. In this context, the number of biosimilars being studied is legally correct. In addition, the three biosimilars are available for their clinical application independently of who manufactures them.
We have updated this information in the revised manuscript in the Materials and methods section.
- With respect to the last question: Given we know that they come from the same manufacturer, how should this be interpreted? In this point, there are two options: (1) they exhibit significant differences or (2) they are not exhibit significant differences. In a such scenario, we are assumed that both products were manufactured using the same manufacturing process but we have not information about their level of variability in order to establish a significance difference. This is an industrial secret.
We need more information to clarify this issue.
-There is a lack of accuracy throughout the text. For instance, in the title “Assessing Analytical Similarity of Three Bevacizumab Biosimilar Marketed in Spain to Avastin®” Biosimilarity is on the product not on the active substance. Hence the expression “bevacizumab biosimilar..to avastin” is not accurate. Also, together with those inaccuracies, the way the manuscript is written makes it difficult to follow. For example, the Conclusion section should be re-arranged. The final sentence does not seem right; after having drawn the conclusions how can the authors state “Finally, the methodology proposed in this study may be adopted by the manufacturer as a component of an integrated control strategy. Furthermore, data derived from stability studies could also be employed to define the specification limits. This topic will be the focus of future research.” (notwithstanding the fact that it does not make much sense to start with “Finally” and follow it by “furthermore”). Those formal mistakes need correction.
Thank for your suggestion and comments. The title and Conclusions section were changed.
-It is unclear what the purpose of the manuscript is: to analyze bevacizumab bearing versions or the decide on the best analytical approach to compare two biologics meant to be the same? (see below comment under Results and Discussion). Although the authors attempt to clarify that in the Intro (3 objectives) it is confusing and the manuscript appears quite unfocused.
We believe that the aims of our study are very clear, although the text appears to be somewhat confusing. We have improved this paragraph in the revised manuscript.
Specific Comments
Title
As this Reviewer understands it, analytical similarity had already been assessed by EMA and was considered proven. What is being shown here is that such similarity is preserved during the early commercial life cycle. Maybe something like “Monitoring analytical similarity between marketed biosimilars bearing bevacizumab and the originator product” would better reflect the contents.
Thanks for your suggestion. The title was changed. We propose the following title: Analytical similarity between three biosimilars bearing bevacizumab marketed in Spain and the originator product.
Abstract
- English needs revision by a native speaker. The text was carefully revised by a native speaker.
- The explanations in the Abstract require further fine-tuning. It seems as if the authors had not paid sufficient attention to make sure that the ideas that they wish to convey are properly delivered. As an example “…in a second step, analytical biosimilarity was assessed using…” Of course I understand what is meant/implied. However, biosimilarity in the abstract is brought into the picture without having mentioned up to that point that biosimilars bearing bevacizumab were also being studied (only reference to the originator had been made). This does not sound right.
- Also the final sentence “Thus, the manufacturing process continue under statistical control. The similarity is demonstrated for the bevacizumab content, independently of the applied approaches, whereas for the dimer content, it is only demonstrated for the biosimilar #1.” It looks like after a conclusive sentence (Thus, the manufacturing process continue under statistical control.) comes a new piece of information (whereas for the dimer content, it is only demonstrated for the biosimilar #1) which is not a proper connection of ideas in this Reviewer’s opinion.
Thanks for your comments. It is true that the explanations in the abstract need more attention in order to give a clear idea of the scope of our study. The abstract was changed.
Introduction
- Please rephrase “With regard to the final product, the biosimilar must be administered in the same manner and via the same route as the reference medicine” given that this is fully accurate; we have a SQ Remsima® (infliximab) that is being called biosimilar of the IV Remicade® (it is unfortunate, but EMA calls the SQ variant a biosimilar)
Thank you for this information. I really think the same about this issue.
The phrase was modified. The biosimilar can be administered in the same manner and via the same route as the reference medicine, although the subcutaneous infliximab biosimilar “Remsima SQ” is an exception.
- The introduction should clearly explain the background and the hypothesis justifying the study undertaken. For instance, the authors refer in the intro to quality attributes such as deamidation or oxidation as well as glycosylation. Why don’t they focus on those attributes.
In accordance with reviewer’s comment, we have changed this paragraph.
Methodology
- I am not sure that Avastin launched in Spain is from Genentech as said. Please verify (in any case Genentech misses an “n”).
Genentech (the name is OK), a Roche subsidiary, developed Avastin® bevacizumab for oncological indications. In Europe, Genentech has licensed the commercialization rights to Avastin® to Roche. In this context, Avastin® batches currently come from Roche, Germany, whereas the first six batches were provided by Roche from London.
- Can authors draw any conclusion on the level of similarity with only 1 batch analysed?
We only have data from one batch for each biosimilar product and it is impossible to draw conclusions. In such a scenario, simulation is a good tool to analyze and learn theoretical models when real data are not enough and not available to study any problem. In the bibliography there are many examples, even we have used this approach in various published works. The conclusions of the simulation study need to be verified with real data. Our study is limited by the availability of biosimilar product batches, only two per year (information provided by the laboratory). For this reason, our study covers a period of three years in order to have the number of batches necessary to apply the QR using real data.
At the end of December or the beginning of January, a second batch will be made available for each of the biosimilars.
- It is not clear from the M&M section what quality attributes are being analyzed. This section if by all means not properly conveyed and a thorough rewriting is advised. What attributes are studied? Why those and how? Etc . One reads that section and does not obtain a clear idea of what is being analyzed.
Thank you for suggesting this; it's true, it is necessary to add a brief description of the quality attributes used in this study. In accordance with reviewer’s comment, we provide additional information about this issue.
Results and discussion
- The discussion seems to involve rather whether a comparison between population means or between populations (QR method) is more appropriate. So, it looks like bevacizumab is not really the subject, but this specific medicine is used as tool to deepen into that discussion. Discussion is unfocused and unorganized in this reviewer’s opinion.
This section was revised and reorganized to facilitate its following. Bevacizumab has been used as a model for the analysis of methodologies and QR approaches in the comparison. This is expected to be applied to other biosimilar studies.
- Submission of a new manuscript that deals with just one of the objectives is advisable.
The reviewer does not explicitly state the study's objective. A similar situation was observed when the reviewer says that “the paper would have been of relevance had the study been executed differently”. The reviewer doesn't say how this could be done. So, it is very difficult for these authors to provide an alternative text.
The study had three principal objectives: the first two relate to the manufacturing process for the reference products and the assessment of its control, while the third is the evaluation of analytical similarity through the application of two methods based on the Quality Range (QR) approach. For this, the originator product (Avastin®) and three authorized biosimilar in Spain since 2021 were used. In accordance with reviewer’s comment, this presupposes the separation of the original manuscript into two parts. One of these may be difficult to fit into the scope of the biosimilars topic. In addition, we need to know the mean and the standard deviation of the reference product to apply the QR approach.
We believe that the division of the article into two parts is unnecessary and have therefore decided to keep it as one piece of content. The data used allows the application and demonstration of the advantages and benefits of the methodology presented, which is part of the novelty of this work. However, we will implement the changes recommended by the reviewer to improve the article in line with their comments and suggestions.
Reviewer 2 Report
Comments and Suggestions for Authors
This manuscript assessed analytical similarity of three bevacizumab biosimilar products in Spain to Avastin, the reference product by using two methods, the FDA approved QR method and the proposed QRML method. The QRML method accounted for between- and within-batch variability and was advantageous over the QR method. Specific comments:
1. Line 171-173: if a total of 10 batches were used in the current study, the sum of the batches described was not 10. Please clarify. “In a first study” may be revised to avoid confusion.
2. Line 177: ‘2.09’, was it a typo? It should be 0.209 according to Table 1.
3. Line 181: ‘six batches’, please clarify if the batches and data were from reference 30. Please cite the reference as needed.
4. Table 1: please clarify if the data of batch 1 to batch 6 were from reference 30. Please add the reference as needed.
5. Table 2: please clarify what were the first six batches and the 10 batches. The authors mentioned six and ten batches in many other places throughout the manuscript, such as in Figure 1 (line 226), Table 3, and Table 4. Please clarify.
6. Line 202: please delete ‘the second’ and clarify the CQA was content to avoid confusion.
Author Response
Reviewer #2
This manuscript assessed analytical similarity of three bevacizumab biosimilar products in Spain to Avastin, the reference product by using two methods, the FDA approved QR method and the proposed QRML method. The QRML method accounted for between-and-within-batch variability and was advantageous over the QR method. Specific comments:
- Line 171-173: If a total of 10 batches were used in the current study, the sum of batches described was not 10. Please clarify. “In a first study “may be revised to avoid confusion.
Thanks for your suggestions. The text was revised to avoid confusion.
- Line 177 “2.09”, was it a typo? It should be 0.209 according to Table 1.
It´s true. The data was corrected.
- Line 181: “six batches”, please clarify if the batches and data were from reference 30. Please cite the reference as needed.
This information was added in the text.
- Table 1. Please clarify if the data of batch 1 to 6 were from reference 30. Please add the reference as needed.
This issue has been revised in all tables (1 to 4) and throughout the text, including Figure 1.
- Table 2: please clarify what were the first six batches and the 10 batches. The authors mentioned six and ten batches in many places throughout the manuscript, such as in Figure 1 (line 226), Table 3 and Table 4. Please clarify.
This issue has been revised in all tables (1 to 4) and throughout the text, including Figure 1.
- Line 202: please delete “the second” and clarify the CQA was content to avoid confusion.
Thank for your suggestion. The text was revised and the CQA was mentioned to avoid confusion.
Reviewer 3 Report
Comments and Suggestions for Authors
The manuscript presented by Oliva et al., deals with the analytical similarity of the biosimilar products. The study was carried out with data from 10 years. The analytical biosimilarity was evaluated by two methods, one proposed by FDA and other by EMA. The topic is very interesting and relevant. It’s very complex to analyze biosimilarity, even the regulatory agencies (FDA, EMA) are not in consensus. This can be mostly attributed to the manufacturing processes, that is an industrial secret.
In my opinion the manuscript can be accepted for publication. I have three minor comments:
Line 25. Specify QRML abbreviation.
The supplementary materials were not available.
I suggest including a representative chromatogram in supplementary material.
Author Response
Reviewer #3
The manuscript presented by Oliva et al., deals with the analytical similarity of the biosimilar products. The study was carried out with data from 10 years. The analytical biosimilarity was evaluated by two methods, one proposed by FDA and other by EMA. The topic is very interesting and relevant. It’s very complex to analyze biosimilarity, even the regulatory agencies (FDA, EMA) are not in consensus. This can be mostly attributed to the manufacturing processes, that is an industrial secret.
In my opinion the manuscript can be accepted for publication. I have three minor comments:
Line 25. Specify QRML abbreviation.
The supplementary materials were not available.
I suggest including a representative chromatogram in supplementary material
Thank you very much for your comments. The abbreviation QRML was included in the text. Supplementary material has been provided up to the submission deadline, but will only be available if the manuscript is published.